# NMR-Solver: automated structure elucidation via large-scale spectral matching and physics-guided fragment optimization

Yongqi Jin [1,2], Jun-Jie Wang[2,3], Fanjie Xu [2,4], Xiaohong Ji[2], Zhifeng Gao [2], Linfeng Zhang[2,5], Guolin Ke [2] ✉, Rong Zhu [3,5] ✉ & Weinan E[1,5,6] ✉

Nuclear Magnetic Resonance (NMR) spectroscopy is one of the most powerful and widely used tools for molecular structure elucidation in organic chemistry. However, the interpretation of NMR spectra to determine unknown molecular structures remains a labor-intensive and expertise-dependent process, particularly for complex or novel compounds. Although recent methods have been proposed for molecular structure elucidation, they often underperform in real-world applications due to inherent algorithmic limitations and limited high-quality data. Here, we present NMR-Solver, a practical and interpretable framework for the automated determination of small organic molecule structures from $^1H$ and $^{13}C$ NMR spectra. Our method introduces an automated framework for molecular structure elucidation, integrating large-scale spectral matching with physics-guided molecular optimization that exploits atomic-level structure–spectrum relationships in NMR. We evaluate NMR-Solver on simulated benchmarks, curated experimental data from the literature, and real-world experiments, demonstrating its strong generalization, robustness, and practical utility in real-life scenarios. By integrating computational NMR analysis, deep learning, and interpretable chemical reasoning into a unified system, it facilitates scalable, automated, and chemically meaningful molecular structure elucidation, establishing a generalizable paradigm for solving inverse problems in molecular science.

Nuclear magnetic resonance (NMR) spectroscopy stands as the most informative and widely used technique for molecular structure elucidation in organic chemistry[1]. Unlike mass spectrometry or infrared spectroscopy, which primarily yield fragmentation patterns or functional groups, NMR provides atomic-level insights into molecular connectivity, stereochemistry, and spatial arrangement[2]. These characteristics make NMR indispensable for the structural characterization of both known and novel small organic molecules, delivering a comprehensive view of the complete molecular architecture.

Despite its analytical power, the interpretation of NMR spectra is a time-consuming and expertise-dependent process. Although computational techniques[3–6] and simulation software[7–9] have made advances in assisting peak picking, shift assignment, and structure validation, manual interpretation remains the standard practice in many laboratories. This reliance on human expertise not only limits throughput but also introduces variability and potential errors, particularly for complex or unknown compounds. Compounding this challenge is the vastness of chemical space—estimated to encompass over $10^{60}$ plausible organic molecules[10]—which makes structure elucidation a highly

[1]School of Mathematical Sciences, Peking University, Beijing, China. [2]DP Technology, Beijing, China. [3]College of Chemistry and Molecular Engineering, Peking University, Beijing, China. [4]Institute of Artificial Intelligence, Xiamen University, Xiamen, China. [5]AI for Science Institute, Beijing, China. [6]Center for Machine Learning Research, Peking University, Beijing, China. ✉e-mail: kegl@dp.tech; rongzhu@pku.edu.cn; weinan@math.pku.edu.cn

complex task. As high-throughput experimentation and AI-assisted automated synthesis platforms continue to advance[11-16], automated spectral analysis is playing an increasingly crucial role in driving the discovery of synthetic pathways and new reactions[17-19]. This, in turn, has fueled an escalating demand for rapid, accurate, and automated NMR-based structure elucidation to keep pace with the accelerating advancements in synthetic chemistry.

Recent advances in deep learning have significantly improved the accuracy and efficiency of predicting chemical shifts from known molecular structures—a key component of forward NMR simulation[20-23]. Both 2D graph-based models and 3D conformation-aware architectures now achieve accurate predictions comparable to density functional theory (DFT)[24] calculations, with orders of magnitude faster inference, exemplified by GT-NMR[25] and NMRNet[26], respectively. These capabilities have enabled large-scale spectral matching and provide a foundation for addressing the more challenging inverse problem: determining molecular structures from experimental NMR spectra.

In parallel with advances in forward spectral modeling, several approaches have been proposed to address this inverse problem, broadly falling into two categories—AI generative models and traditional approaches. Among the recent generative modeling approaches, the typical framework is encoding NMR spectra into token sequences and employs a sequence-to-sequence architecture to generate molecular text representations[27-29], such as SMILES[30] and SELFIES[31]. While these end-to-end frameworks eliminate the need for manual feature engineering, they suffer from limited generalization and poor interpretability. Moreover, due to the scarcity of high-quality experimental datasets, such models are typically trained on simulated data, leading to a significant domain gap between training and real-world conditions. This undermines their reliability in practical applications and their ability to generalize to out-of-distribution chemical structures.

In contrast, traditional optimization strategies—such as genetic algorithms[32,33]—offer greater transparency by explicitly exploring molecular space through iterative refinement[34,35]. To obtain a suitable initial population, database lookups are commonly employed to rapidly select from known compounds[36-38]. However, their stochastic and undirected search mechanisms lead to inefficient and unstable navigation of the vast chemical space, often failing to converge to the desired molecules within a feasible computational time. These challenges are further exacerbated when addressing the objective of structure elucidation, where only a single, definitive solution exists.

Despite their differing paradigms, both generative and traditional approaches fail to reliably bridge NMR spectra with novel, chemically valid structures under practical conditions—either due to poor reliability and generalization, or inefficiency arising from random exploration. There remains a critical need for more effective strategies that fully leverage NMR spectral information and demonstrate robustness and reliability in practical applications.

To address these challenges, we propose NMR-Solver, an automated and interpretable framework for determining small organic molecule structures from $^1H$ and $^{13}C$ NMR spectra. Built on a physics-guided, fragment-based optimization strategy, NMR-Solver integrates large-scale spectral matching with atomic-level structure–spectrum correlations to guide molecular assembly in a chemically meaningful and transparent manner. Unlike traditional approaches that rely on random and undirected search, our method navigates chemical space in a targeted manner by evolving molecular fragments based on observable spectral features. This enables robust and efficient structure elucidation in real-world scenarios—where novel scaffolds, spectral ambiguities, and incomplete data pose significant challenges—while ensuring full traceability of each structural decision to experimental evidence. As a result, NMR-Solver provides a reliable and trustworthy solution for practical structure determination.

## Results

### Overview of the NMR-Solver framework

As illustrated in Fig. 1, the NMR-Solver framework consists of four core modules: molecular optimization, forward prediction, database retrieval, and scenario adaptation. Operating in a closed-loop manner, it addresses the inverse NMR problem by starting from initial candidate structures—obtained via database retrieval or user input—and iteratively refining them through fragment-based molecular optimization. In each iteration, the spectra of evolving candidates are simulated and compared against experimental data, guiding the search

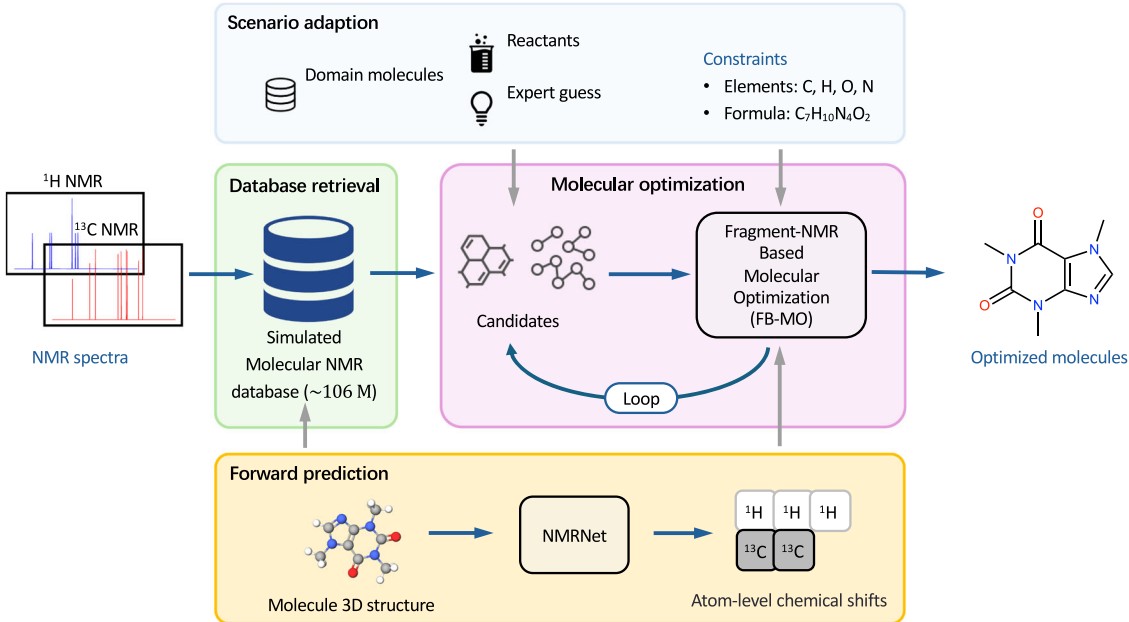

**Fig. 1 | Architecture of NMR-Solver.** The framework operates in a closed loop and integrates four modules: (i) forward prediction of $^1H$ and $^{13}C$ chemical shifts using NMRNet; (ii) large-scale database search over 106 million entries; (iii) Fragment-NMR-Based Molecular Optimization (FB-MO) for iterative refinement; and (iv) a scenario adaptation interface allowing user-defined candidates and constraints.

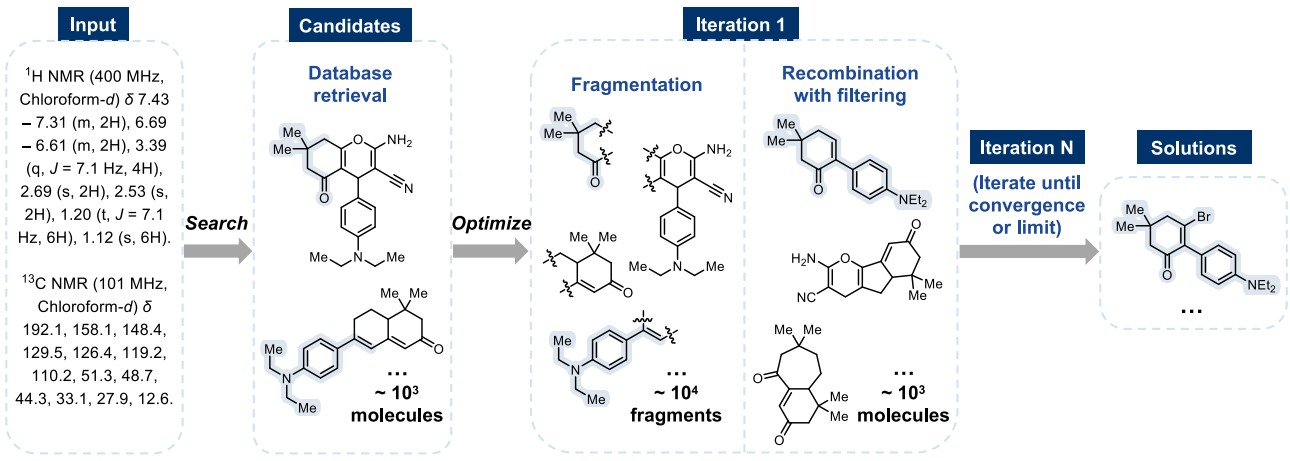

**Fig. 2 | Workflow of NMR-Solver.** Starting from experimental NMR data, candidate molecules are retrieved from a large database. They are fragmented and recombined in an iterative loop, with each iteration filtering new candidates by spectral matching using predicted chemical shifts. The process continues until convergence, producing chemically valid and spectrally consistent solutions.

toward chemically valid and spectrally consistent solutions (see Fig. 2 for an illustration of the iterative refinement process).

Central to NMR-Solver is the Fragment-NMR-Based Molecular Optimization (FB-MO) module, which employs a physics-guided, fragment-based strategy to iteratively refine candidate structures. In a single iteration, the module generates new candidates by considering chemically valid fragment recombinations from the current molecular pool, potentially exceeding $10^9$ possibilities under typical settings. Instead of exploring this space randomly, the search is guided by atomic-level structure–spectrum correlations, enabling efficient filtering of potential offspring whose simulated spectra closely match the experimental data. This focused, evidence-driven exploration ensures both high efficiency and interpretability, with each structural optimization traceable to specific NMR features.

To support rapid spectral evaluation during optimization, we employ pre-trained NMRNet[26], an SE(3)-equivariant Transformer[39] architecture that predicts $^1$H and $^{13}$C chemical shifts with high accuracy (reported MAE: 0.181 ppm for $^1$H, 1.098 ppm for $^{13}$C), comparable to DFT calculations. Crucially, NMRNet achieves this predictive performance at a computational cost several orders of magnitude lower than DFT, enabling real-time spectral simulation during iterative optimization and large-scale database construction.

For efficient candidate initialization, the framework leverages a large-scale spectral retrieval module that identifies plausible molecular candidates by querying the SimNMR-PubChem Database, which we constructed from approximately 106 million small organic molecules sourced from PubChem[40]. Each molecule in this database is annotated with predicted chemical shifts, making it the largest simulated NMR database available to date. This substantially exceeds the scale of existing publicly available datasets, such as NMRShiftDB2[41], NP-MRD[42], QM9-NMR[43], and Multimodal Spectroscopic Dataset[44]. By leveraging modern vector database techniques—including approximate nearest neighbor (ANN) search—and applying re-ranking strategies, the database retrieves spectrally similar candidates in sub-second time, providing high-quality starting points for molecular optimization.

Finally, the scenario adaptation module integrates domain-specific knowledge—such as known reactants or proposed scaffolds—as initial inputs for guided structural search. It also allows the application of constraints on molecular formula and elemental composition during optimization and filtering. These capabilities together enable flexible deployment across a range of scenarios, from ab initio structure elucidation to reaction-aware product inference, supporting both automated analysis and human-in-the-loop interpretation.

## Structure elucidation on simulated spectra

Due to the scarcity of large-scale, high-quality experimental NMR datasets with carefully curated structure–spectrum pairs, most existing works are evaluated on simulated datasets. To facilitate a fair comparison with previous methods, we evaluated NMR-Solver on the publicly available simulated benchmark dataset introduced by Alberts et al.[29]. This dataset has been used to benchmark several state-of-the-art approaches, including NMR-to-Structure[29] and GraphGA with Multimodal Embeddings[35], whose performance results are reported in the original study. We compare our method directly against these approaches, enabling a meaningful assessment of NMR-Solver's capabilities within the context of current state-of-the-art techniques.

The simulated dataset comprises 345,000 SMILES-spectrum pairs, with $^1$H and $^{13}$C NMR spectra simulated using MestReNova[8]. In the $^1$H spectra, each peak includes chemical shifts, peak integrations, $J$-coupling constants, and multiplicity patterns, while the $^{13}$C spectra contain only chemical shifts. Although this dataset is frequently used for training and evaluation, it introduces a significant gap when compared to real experimental conditions. Idealized $^1$H multiplicity patterns and $J$-coupling values, which are often not fully or accurately observable in actual experiments due to factors like solvent effects, peak overlap, and instrument resolution, may lead previous methods to overfit these features in the simulated $^1$H NMR spectra. To mitigate these potential biases, we evaluated NMR-Solver without using $J$-coupling or multiplicity information, relying solely on chemical shifts and peak integrations—features that are more reliably measurable in practice. Due to computational resource limitations, we evaluate the method on a test set by randomly sampling 1000 molecules from the dataset.

We evaluated the method using the original metrics, requiring an exact molecular match with stereochemistry considered, and the results are summarized in Table 1. Although our approach does not use the same simulation pipeline as GraphGA with Multimodal Embeddings (MestReNova) or rely on the benchmark dataset like NMR-to-Structure, it still achieves performance comparable to state-of-the-art methods on both $^{13}$C-only and combined $^1$H/$^{13}$C settings. The performance gap in the $^1$H-only setting occurs due to previous methods' dependence on idealized $^1$H multiplicity patterns and $J$-coupling values from simulated spectra.

This indicates that NMR-Solver effectively leverages experimentally robust spectral features for accurate molecular structure prediction, rather than relying on fragile information such as idealized multiplicity patterns or artificially noise-free simulations. In the following section, we further evaluate the framework on experimentally

**Table 1 | Comparison with current methods on the simulated dataset**

| Input spectra | Conditions | Top-1 (%) | Top-5 (%) | Top-10 (%) |
|---|---|---|---|---|
| **GraphGA with Multimodal Embeddings[35]** | | | | |
| $^1$H NMR + $^{13}$C NMR + IR | Formula | (76.02) | (87.81) | (88.91) |
| **NMR-to-Structure[29]** | | | | |
| $^1$H NMR | Formula | **62.70** (55.32) | **75.40** (73.59) | **80.80** (76.74) |
| $^{13}$C NMR | Formula | 48.40 (53.91) | 61.70 (73.45) | 70.10 (77.72) |
| $^1$H NMR + $^{13}$C NMR | Formula | **69.60** (66.99) | 81.70 (84.09) | 86.80 (86.59) |
| **NMR-Solver (Ours)** | | | | |
| $^1$H NMR | Formula | 23.10 | 36.20 | 39.30 |
| $^{13}$C NMR | Formula | **62.20** | **77.60** | **79.10** |
| $^1$H NMR + $^{13}$C NMR | Formula | 66.90 | **87.20** | **89.90** |

The performance gap in the $^1$H-only setting occurs due to previous methods' dependence on idealized simulated $^1$H multiplicity patterns and *J*-coupling values. All methods are re-implemented and tested on 1000 randomly sampled molecules. Bold results indicate the best performance. Numbers in parentheses indicate the results directly reported in the original publications under their respective evaluation settings, and are provided for reference only. "Conditions: Formula" refers to the known molecular formula of the target molecule, i.e., the number of atoms of each element.

measured NMR spectra curated from the literature, highlighting its generalization capability and practical robustness in real-world structure elucidation tasks.

## Generalization to experimental spectra

To enable realistic and rigorous benchmarking of structure elucidation methods, we manually curated a dataset of approximately 450 reactant–product pairs with product NMR spectra from original research articles published in the *Journal of the American Chemical Society* in 2024.

The data were extracted from 15 weekly issues, covering a broad range of organic chemistry disciplines, total synthesis, catalysis, and substrate preparations in polymer chemistry and biochemistry. For each article, 3–5 representative reactions were selected to maximize structural and functional diversity. Each entry includes the key reactants, the reported product structure, and its corresponding experimental $^1$H and $^{13}$C NMR spectra. Minor manual curation was performed to remove peaks originating from solvents or impurities, ensuring high data quality. This benchmark reflects real-world conditions in synthetic chemistry and provides a robust foundation for evaluating computational structure elucidation approaches.

A simple statistic highlights the limitations of existing simulated benchmarks discussed in the previous subsection. In the literature-extracted dataset, over 40% of $^1$H NMR peaks are annotated as "m" (multiplet), without interpretable coupling constants. In contrast, the simulated dataset used by previous methods features fully resolved multiplicities and *J*-coupling values of $^1$H NMR peaks, which do not account for the complexities of real-world measurements.

We evaluated NMR-Solver and NMR-to-Structure using both recall and Tanimoto similarity[45] computed on Morgan fingerprints[46], which measure exact structure recovery and substructural similarity between predicted and ground-truth molecules, respectively. As shown in Fig. 3a, the previous method, NMR-to-Structure, exhibits limited performance, achieving only 14.44% (top-1) and 21.78% (top-10) accuracy when using $^1$H and $^{13}$C NMR spectra under the given molecular formula, while NMR-Solver substantially outperforms these baselines, achieving 52.89% (top-1) and 67.33% (top-10) recall under the same evaluation conditions. Moreover, as shown in Fig. 3b, even for predictions that do not exactly match the ground truth, NMR-Solver yields structures with

higher Tanimoto similarity, reflecting improved substructural consistency and chemically meaningful outputs. A complete breakdown of experimental results is provided in Supplementary Tables 4 and 5.

These results highlight a substantial performance gap when models trained on simulated data are applied to real-world, experimentally measured NMR spectra. In contrast, our method does not rely on large-scale simulated data for training, but instead leverages simulation only during the scoring stage, leading to greater robustness in practical scenarios.

Furthermore, NMR-Solver ranks candidate structures by spectral similarity between simulated and experimental spectra. As illustrated in Fig. 3c, prediction accuracy improves monotonically with increasing spectral similarity, suggesting that the spectral match score can serve as a reliable confidence indicator for the predictions. This self-consistent scoring mechanism enhances the reliability and practical applicability of NMR-Solver in real-world structure elucidation tasks.

## Product structure prediction guided by reaction context

In practical structural analysis, product structure prediction is a common and significant scenario, where chemists interpret NMR spectra in the context of known reactants or anticipated chemical transformations. To evaluate NMR-Solver's ability to leverage such prior knowledge, we incorporated reactant information from the literature-extracted dataset described above.

By incorporating reactant structures into the initial candidate pool, NMR-Solver leverages substructures probable to be retained in the product and reflected in the NMR spectra. This allows the method to effectively bias molecular optimization toward candidates that contain spectrally supported, chemically plausible fragments.

As illustrated in Fig. 3d, this strategy increases the top-1 recall from 52.89 to 60.22% and the top-10 recall from 67.33 to 76.22% under $^1$H and $^{13}$C NMR conditions with the molecular formula provided, demonstrating that leveraging structural continuity across reactions enhances inference accuracy.

These results highlight how NMR-Solver combines automated analysis with human-like reasoning: rather than operating in isolation, it amplifies expert knowledge and integrates reaction information in a flexible and robust manner, improving both the reliability and general applicability of molecular structure elucidation in synthetic workflows. While our evaluation focuses on incorporating reactant structures, the framework is generalizable and can similarly utilize expert hypotheses or domain-specific molecular priors to guide optimization when available.

## Evolution of spectral similarity during molecular optimization

During the molecule optimization process, the agreement between predicted and experimental NMR spectra evolves as candidate molecules are iteratively refined. As shown in Fig. 3e, f, spectral similarity scores for top candidates increase substantially within the first few optimization steps, indicating rapid convergence toward structures with high spectral agreement.

Specifically, the median spectral similarity of the top-10 candidates surpasses 0.96 within the first two iterations—a threshold empirically associated with 50% top-10 prediction accuracy—and gradually approaches a plateau of 0.97. The diminishing gains in later stages reflect that the search quickly homes in on chemically meaningful structures.

This rapid progression highlights the efficiency of NMR-Solver in navigating the vast chemical space: by prioritizing molecular structures that better explain the observed spectra, the method identifies high-quality candidates early, without the need for exhaustive exploration. In contrast, conventional approaches, such as genetic algorithms, rely on random crossover and mutation operations, lacking directed guidance. These methods explore chemical space indiscriminately, often requiring many more iterations to converge and producing candidates of highly variable quality.

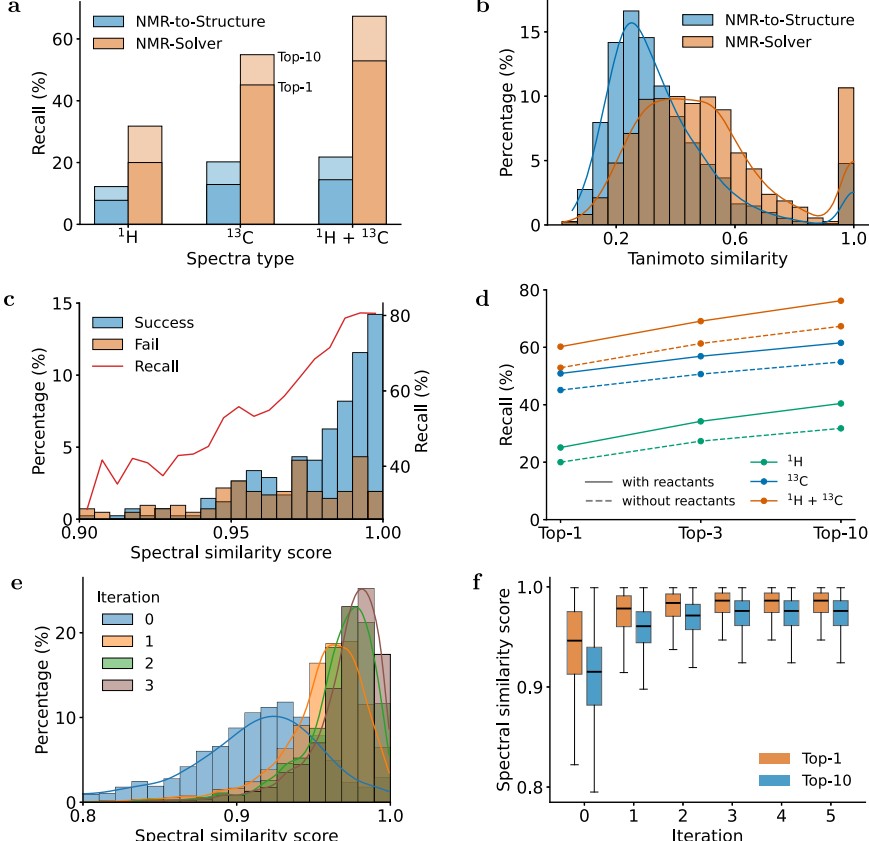

**Fig. 3 | Performance of structure elucidation on literature data. a** Comparison of top-1 and top-10 recall for different methods under the formula condition. **b** Comparison of tanimoto similarity distributions for top-10 predictions, excluding invalid predictions. **c** Relationship between average top-10 spectrum similarity score and success rate. **b**, **c** Results are conducted by NMR-Solver using ¹H and ¹³C NMR under the formula condition. **d** Comparison of recall rates for NMR-Solver with and without reactants. **e** Dynamics of spectral similarity scores across iterations. **f** Top-1 and top-10 score distributions are shown as box plots. Box plots show the median (center line), interquartile range (box), and 10th–90th percentiles (whiskers), $n = 450$ samples.

## Real-world experimental validation

To further assess the practical utility of NMR-Solver in real-world synthetic chemistry research, we applied it to challenging cases from laboratory experiments—situations where traditional manual analysis is hard to provide conclusive structural assignments. Representative cases from real-world experiments are presented in Fig. 4.

In some instances, reaction pathways are highly complex or unanticipated, making it difficult for chemists to hypothesize plausible product structures based on mechanistic reasoning alone; without such hypotheses, manual NMR interpretation becomes extremely challenging. An illustrative example (Fig. 4a) from our laboratory involved a palladium-catalyzed transformation. The ¹H and ¹³C NMR spectra of the product suggested the presence of two Ph groups, one $CO_2Et$ unit, and a tri-substituted benzene. However, based on these spectral features and considering possible catalytic pathways, we were unable to propose a reasonable product structure, and the study was temporarily halted. Applying NMR-Solver to the same NMR data enabled the identification of a candidate structure that consistently matched the observed chemical shifts and coupling patterns. Based on this structure, we were able to propose a reasonable reaction mechanism (Fig. 4a, bottom). Subsequent validation via two-dimensional NMR experiments (HMBC and HSQC) and high-resolution mass spectrometry (HRMS) confirmed the correctness of the assignment.

Unexpected side products pose additional challenges. For instance, in an attempt to convert a hydroxyl group into a sulfonate ester (Fig. 4b), chromatographic isolation yielded a byproduct lacking any aromatic protons. Initial hypotheses considered included chlorination of the hydroxyl group, which would introduce one additional

proton compared with the observed spectrum, or the formation of an eight-membered cyclic byproduct; however, HRMS analysis showed no mass corresponding to the eight-membered cyclic structure. When NMR-Solver was applied, it confidently proposed a dichlorinated product whose predicted spectrum matched the experimental data, and subsequent HRMS confirmed the molecular formula of the dichloride, validating this assignment.

Isomeric products further complicate structural elucidation, as many isomers produce very similar ¹H and ¹³C spectra, making them difficult to distinguish without additional references. For example, in the case of hydrofunctionalization of cinnamate esters (Fig. 4c), two regioisomers are possible[47]. Manual interpretation of the one-dimensional spectra without references makes it difficult to distinguish the products. NMR-Solver, however, accurately discriminated between the isomers based solely on the one-dimensional spectra, correctly identifying the site of hydrogenation and thereby corroborating the proposed reaction mechanism.

Beyond facilitating elucidation of previously unassignable compounds, NMR-Solver can detect and correct misassigned structures in the literature. Two cases (Fig. 4d, e) originally reported by Fu et al.[48] and Cheng et al.[49], and later corrected by Novitskiy et al.[50], were re-analyzed with NMR-Solver. The corrected structures were accurately predicted and received substantially higher spectral similarity scores than the originally proposed ones, demonstrating that NMR-Solver can independently identify inconsistencies that might elude manual interpretation, even in peer-reviewed publications.

A key limitation of NMR-Solver lies in its reliance on the accuracy of the underlying forward prediction model. In one example involving

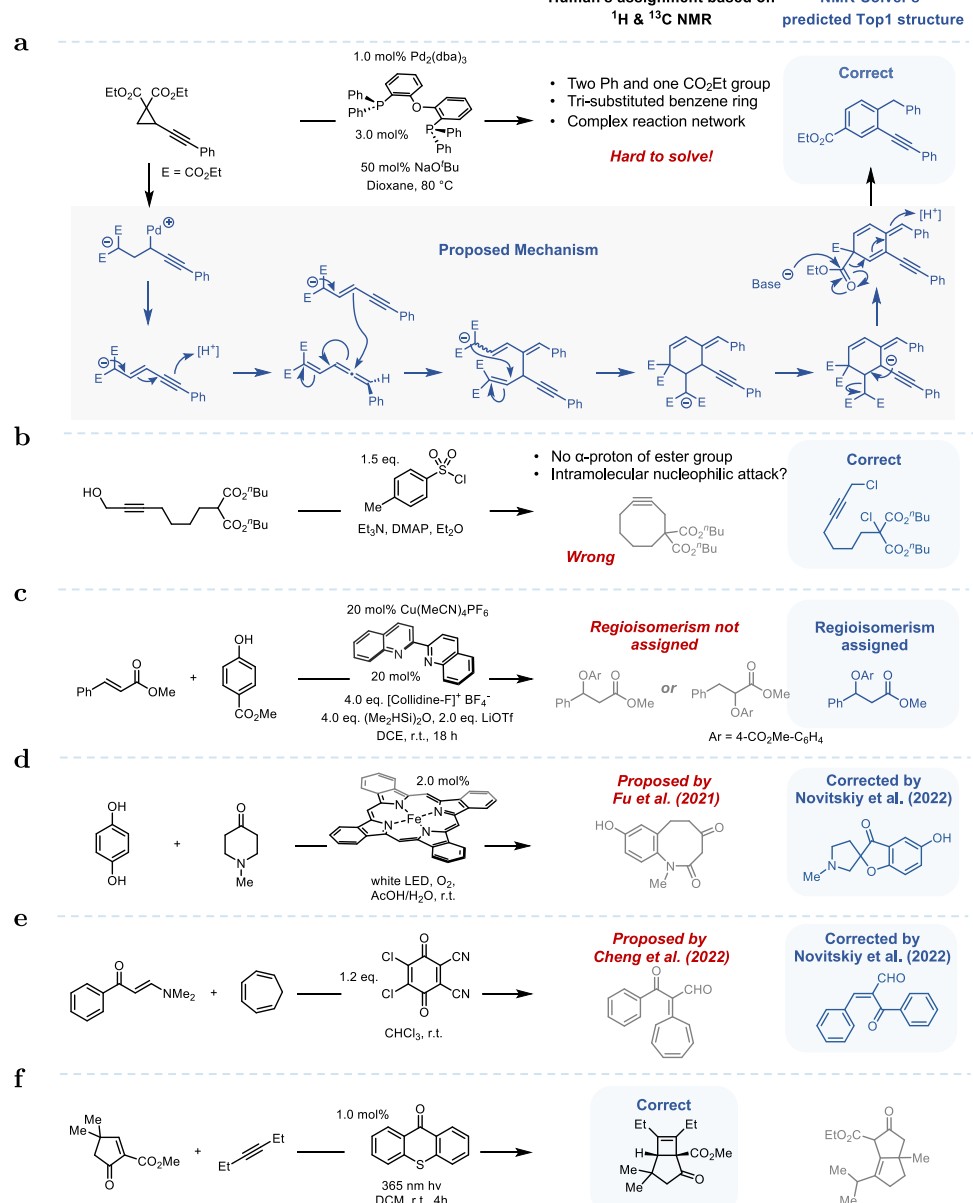

**Fig. 4 | Real-world experimental cases demonstrating the utility of NMR-Solver.** **a** Challenging laboratory case where manual NMR analysis failed, but NMR-Solver correctly predicted the structure. **b** Unanticipated side product lacking aromatic protons, correctly proposed as a dichlorinated structure by NMR-Solver. **c** Regioisomer discrimination in cinnamate ester hydrofunctionalization, correctly resolved by NMR-Solver. **d**, **e** Correction of two misassigned structures in the literature using NMR-Solver. **f** Failing to predict the correct structure due to inaccuracies in the forward prediction model.

a strained system (Fig. 4f), the prediction model (NMRNet) exhibited a relatively large deviation, leading NMR-Solver to favor a structure with a predicted spectrum closer to the experimental data rather than the real product. Future advances and anticipated improvements in NMR prediction algorithms will be incorporated into NMR-Solver to continuously advance its accuracy in structural determination.

## Discussion

NMR-Solver enables accurate, robust, and interpretable structure elucidation directly from experimental ${}^{1}$H and ${}^{13}$C NMR spectra. Unlike many existing methods that treat structure elucidation as a black-box translation task, NMR-Solver constructs molecules through chemically meaningful steps guided by spectral evidence, making interpretability intrinsic to the generation process. At the same time, it achieves state-of-the-art performance and strong robustness against common

spectral ambiguities, supporting reliable application under realistic experimental scenarios.

Looking ahead, two directions appear particularly important for advancing the field. First, systematic curation of large-scale experimental NMR datasets—from literature mining or high-throughput measurements—will be critical for training and benchmarking methods under practical conditions. Second, continued progress in forward spectral simulation, especially in accurate chemical shift prediction, will further enhance the reliability and generalizability of inference. With its modular design, NMR-Solver is well positioned to leverage improvements in both dataset availability and spectral simulation.

By integrating physics-informed structure–spectrum relationships into the optimization process, NMR-Solver establishes a guided and robust paradigm for molecular optimization, moving beyond the inefficiencies and instability of traditional stochastic approaches. This

framework also holds promise for extension to other spectroscopic modalities and broader molecular design applications.

Finally, as an interpretable and flexible analysis platform, NMR-Solver accommodates both fully automated and human-in-the-loop spectrum interpretation. In the context of emerging automated laboratories, it can serve as a key component for validating reaction outcomes and inferring products, thereby accelerating the discovery of new reactions and synthetic pathways. From a broader perspective, such capabilities underscore its potential role in driving a paradigm shift toward automated scientific research.

## Methods

### Data preparation

NMR spectra are processed into a list of peaks annotated with relevant features. For $^1$H NMR, each peak includes chemical shift, integration (proton count), multiplicity, and $J$-coupling constants; for $^{13}$C NMR, only the chemical shift is retained.

For literature-reported spectra, we extract peak information using regular-expression parsing. For experimentally acquired spectra, peaks are annotated using standard NMR preprocessing procedures. Specifically, the raw FID files are processed using conventional NMR software such as MestReNova[8], including baseline correction, peak picking, integration, and multiplicity assignment.

In experimental $^1$H and $^{13}$C NMR spectra, the chemical shifts of individual peaks are extracted and represented as an unordered multiset, serving as a unified spectral representation. To encode integration information in $^1$H NMR, each chemical shift is repeated in the multiset according to its associated proton count. Multiplicities are additionally attached to each shift. When a chemical shift is reported as a range, the midpoint of the range is taken as the representative value.

### Spectral simulation

Chemical shift predictions are generated using NMRNet, a deep learning model that takes atomic types and three-dimensional coordinates as input and predicts chemical shifts of all $^1$H and $^{13}$C nuclei. Prior to prediction, molecular conformations are generated using the `EmbedMolecule` and `MMFFOptimizeMolecule` functions from the RDKit toolkit[51], which perform conformational sampling and geometry optimization under the Merck Molecular Force Field (MMFF)[52].

The final simulated spectra are also encoded as unordered multisets: $\mathcal{H} = \{h_1, \ldots, h_n\}$ for $^1$H NMR, where each element $h_i$ corresponds to the predicted chemical shift of an individual hydrogen nucleus, and $\mathcal{C} = \{c_1, \ldots, c_m\}$ for $^{13}$C NMR, where each element $c_j$ corresponds to a unique carbon signal, with chemical shifts from symmetry-equivalent carbon nuclei averaged into a single resonance. For $^{13}$C NMR, chemical shifts of symmetry-equivalent carbons are averaged into a single resonance. This representation naturally captures signal intensity through the multiplicity of identical shift values arising from chemically equivalent nuclei, and mimics the peak coalescence observed in experimental spectra.

### Similarity metric of NMR spectra

To enable efficient and robust scoring between two NMR spectra, we introduce two complementary similarity metrics: vector similarity designed for scalable retrieval and high-throughput screening, and set similarity tailored to improve accuracy and robustness in the presence of spectral noise and peak mismatches.

**Vector similarity.** Given an NMR spectrum $\mathcal{X} = \{x_1, \ldots, x_n\}$, we smooth it using Gaussian convolution:

$$g(t) = \sum_{i=1}^{n} \exp\left(-\frac{(t - x_i)^2}{2\sigma^2}\right), \tag{1}$$

where $t$ spans the typical chemical shift range for $^1$H or $^{13}$C NMR. The resulting signal is discretized into a vector representation:

$$\mathbf{v}_X = [g(t_1), g(t_2), \ldots, g(t_{128})], \tag{2}$$

with $\{t_i\}$ denoting 128 uniformly spaced sampling points across the spectral window. The specific parameter settings are provided in Supplementary Table 1.

The encoding vector of the full NMR spectrum is formed by concatenating $^1$H and $^{13}$C NMR representations:

$$\mathbf{v} = [\mathbf{v}_H; \mathbf{v}_C] \in \mathbb{R}^{256}, \tag{3}$$

yielding a fixed-dimensional representation suitable for fast similarity scoring and scalable vector indexing. Spectral similarity is then measured using the Euclidean distance:

$$d_{L^2}(\mathbf{v}, \mathbf{v}') = \| \mathbf{v} - \mathbf{v}' \|_2, \tag{4}$$

providing a computationally efficient metric for high-throughput screening.

**Set similarity.** We define set similarity as the optimal alignment between two collections of chemical shifts, formulated as a bipartite matching problem. Let

$$\mathcal{X} = \{x_1, \ldots, x_m\}, \quad \mathcal{Y} = \{y_1, \ldots, y_n\} \tag{5}$$

denote two spectra. The similarity score is defined as

$$S(\mathcal{X}, \mathcal{Y}) = \frac{1}{\sqrt{mn}} \max_{P \in \mathcal{P}} \sum_{(i,j) \in P} f(x_i, y_j), \tag{6}$$

where $\mathcal{P}$ denotes the set of all one-to-one matchings (allowing unmatched elements), and $f(x, y)$ is a kernel function measuring pairwise compatibility:

$$f(x, y) = \exp\left(-\frac{(x - y)^2}{2\sigma^2}\right). \tag{7}$$

The optimization problem in Eq. (6) can be fast solved using the Kuhn–Munkres algorithm[53,54], as implemented in `linear_sum_assignment` from `scipy.optimize`[55]. In addition, multiplicity patterns can be incorporated by weighting the kernel function, thereby introducing additional spectral information (see Supplementary Note 3 for implementation details).

Compared to vector-based similarity metrics, set-based similarity is more tolerant to missing or noisy peaks, enabling final robust identification of molecules even when spectra partially overlap. This reflects a design that is inherently robust to noise and imperfections in peak extraction under realistic experimental conditions. Additional experimental evidence demonstrating these advantages is provided in Supplementary Note 4.

### Database construction and retrieval

We constructed the SimNMR-PubChem Database, a large-scale NMR molecular repository derived from the PubChem dataset[40], which initially contained approximately 119 million compounds. To ensure chemical relevance, ionized species, molecules lacking both carbon and hydrogen, free radicals, and isotopically labeled compounds were excluded. After deduplication using InChIKeys, the final dataset comprises 106 million unique molecules.

For each molecule, we generated simulated $^1$H and $^{13}$C NMR spectra as described in the subsection "Spectral simulation", and

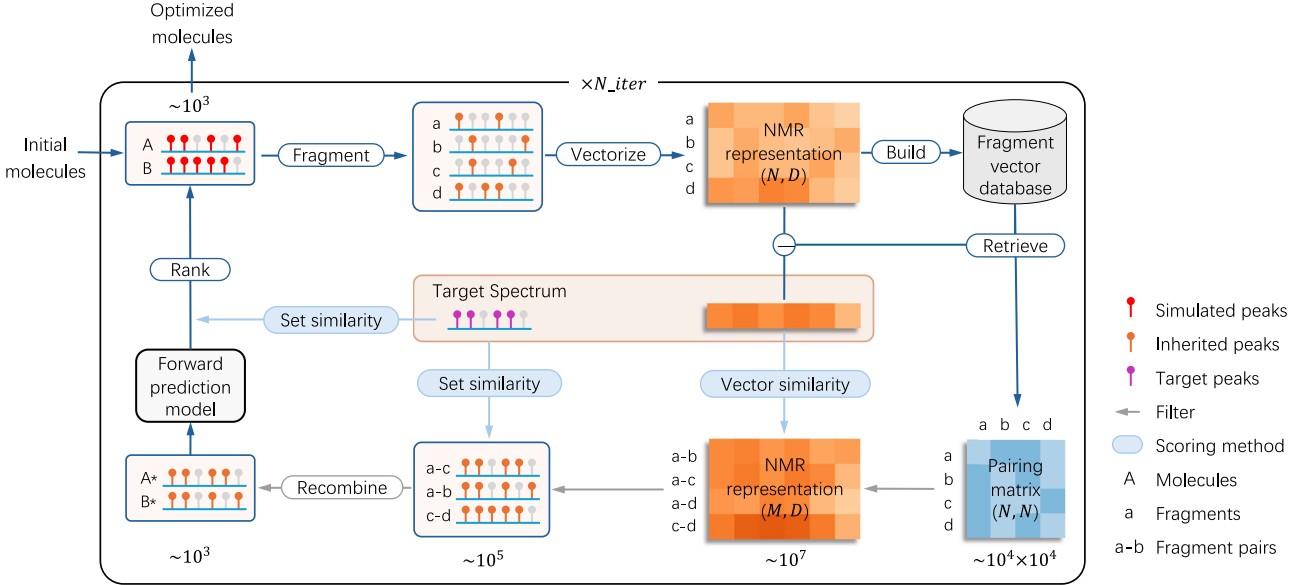

**Fig. 5 | Workflow of Fragment-NMR-Based Molecular Optimization (FB-MO).** Array shapes are indicated in parentheses: $N$, number of fragments; $M$, number of selected fragment pairs; $D$, dimensionality of the vector representation.

subsequently converted them into 256-dimensional encoding vectors as defined in Eq. (3).

To enable scalable similarity search, we implemented a vector database using FAISS[56], employing an HNSW index[57] with cosine similarity for efficient ANN retrieval.

Retrieval is performed in two stages: an initial ANN scan identifies candidate molecules based on vector similarity, followed by re-ranking using the more accurate but computationally slower set similarity. This hybrid approach combines computational efficiency with high-precision spectral matching, enabling sub-second retrieval.

### Fragment-NMR-Based Molecular Optimization

We introduce Fragment-NMR-Based Molecular Optimization (FB-MO), a directed evolutionary strategy that leverages predicted NMR spectra to guide the search toward molecules matching a target spectrum. The overall workflow is summarized in Fig. 5.

The approach maintains a dynamically updated molecule pool, initially populated with compounds retrieved from the database. In each iteration, these molecules are fragmented into structural building blocks and recombined through crossover to generate novel candidates.

To efficiently predict the NMR spectra of newly generated molecules after crossover, FB-MO estimates the chemical shifts of atoms in the new molecule by using the corresponding shifts from the parent molecules. This strategy can be seen as a fragment-based extension of the HOSE (Hierarchically Ordered Spherical Environment) code approach[58], which predicts chemical shifts based on local chemical topology. Its validity is supported by the well-established principle that NMR chemical shifts are primarily governed by local chemical structure, with long-range effects decaying rapidly and typically being negligible[59].

During the molecular fragmentation and recombination process to generate new molecules, atoms at the cleavage sites of the fragments are the only ones whose local environments undergo substantial changes, while the local environments of the remaining atoms stay relatively stable. To preserve these environments, simple constraints are imposed on the recombination based on the cleavage bond, which is defined by the atom types and bond orders at the break site (e.g., C–O, N=C). For carbon-started bonds, such as C–**O**, only

fragments with cleavage bonds starting with the terminal atom (e.g., **O**–C, **O**–N) are allowed. In contrast, for non-carbon-based bonds, like N=C, pairing is allowed with any fragment that shares the same bond order (e.g., single, double, or triple). These constraints ensure that the circular neighborhood of radius 1 for all carbon atoms remains unchanged, preserving the circular neighborhood of radius 2 for most hydrogen atoms. A complete list of permissible cleavage bond pairings is provided in Supplementary Table 10.

Within this framework, the NMR spectrum of a newly formed molecule $M$, assembled from fragments $F_1$ and $F_2$, is estimated by inheriting the chemical shifts of corresponding atoms from their parent molecules:

$$\mathcal{X}_M = \mathcal{X}_{F_1} \cup \mathcal{X}_{F_2}, \tag{8}$$

and the associated encoding vectors are combined additively:

$$\mathbf{v}_M = \mathbf{v}_{F_1} + \mathbf{v}_{F_2}. \tag{9}$$

Candidate selection is performed in three stages to achieve high efficiency and accuracy. First, fragment pairs are pre-screened using a fast vector-based retrieval: for each fragment $F$, the top-$k$ complementary fragments $F'$ are retrieved by minimizing $\| \mathbf{v}_{target} - \mathbf{v}_F - \mathbf{v}_{F'} \|_2$ using an $L^2$ distance-optimized index. Second, all candidate pairs are aggregated and re-ranked by vector similarity. Third, the top candidates are refined using set similarity between $\mathcal{X}_{F_1} \cup \mathcal{X}_{F_2}$ and the target spectrum as given in Eq. (6).

Newly generated molecules undergo precise NMR prediction using the forward model for final scoring. This two-tiered NMR estimation-fast inherited prediction for screening and precise forward prediction for validation-enables both scalability and high fidelity. High-similarity candidates replace lower-ranking molecules in the pool, which evolves iteratively until convergence or the maximum iteration count is reached. A detailed algorithmic description is provided in Supplementary Note 5.

### Additional constraints and prior knowledge integration

To address various experimental constraints and incorporate prior knowledge, the algorithm features several adaptive capabilities. When

the permissible elemental composition is known, fragments containing excluded elements are filtered out at each iteration, which helps reduce computational overhead. If a molecular formula is specified, candidate molecules are then evaluated against this constraint during the final stage.

When reactants are known, they are directly integrated into the initial molecule pool. This is beneficial as reactants often share structural motifs with the target molecule, thereby enhancing both the efficiency and accuracy of the search. Similarly, other types of prior knowledge—such as expert-proposed scaffolds or domain-specific molecules—can be seamlessly incorporated by including them in the initial pool.

## Computational resources and runtime

All experiments were conducted on a single NVIDIA GeForce RTX 4090 GPU. Under the default algorithmic settings, the average wall-clock time per molecule is 94 s when element-based screening is enabled (i.e., when the elements or molecular formula of the target molecule are provided), and 106 s when element-based screening is disabled.

## Reporting summary

Further information on research design is available in the Nature Portfolio Reporting Summary linked to this article.

## Data availability

The PubChem dataset[40], used to construct the SimNMR-PubChem Database, is publicly available at https://pubchem.ncbi.nlm.nih.gov. The processed dataset and database index of the SimNMR-PubChem Database are available on Hugging Face at https://huggingface.co/datasets/yqj01/SimNMR-PubChem. All processed NMR datasets used for testing are available via Zenodo at https://doi.org/10.5281/zenodo.16952024[60]. All datasets generated and analyzed in this study are publicly accessible and can be freely used for research purposes without restriction.

## Code availability

All source code for NMR-Solver is publicly available at https://github.com/YongqiJin/NMR-Solver[61] under the open-source MIT License. The trained model weights for NMRNet are available via Zenodo at https://doi.org/10.5281/zenodo.16952024[60].

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

## Acknowledgements

The authors thank Shangqian Chen and Peng Jin for their contributions to the development of the web app. The authors are also grateful for the insightful discussions and suggestions from Hanzheng Li and Xi Wang. W.E. acknowledges the National Natural Science Foundation of China (grant nos. 92570001 and 12288101). R.Z. acknowledges the New Generation Artificial Intelligence-National Science and Technology Major Project (2025ZD0121905), the National Natural Science Foundation of China (22350006, T2521001, 22222101, 22171012), Beijing Natural Science Foundation (2242006), and the AISI-NUS joint research initiative.

## Author contributions

W.E., R.Z., and G.K. contributed to the design of the work. Y.J. designed and implemented the methods and conducted the analysis. Y.J. and J.W. performed data collection and preprocessing. J.W. conducted the wet-lab experiments. Y.J. and F.X. carried out the evaluation of the methods. X.J., Z.G., and L.Z. contributed to project coordination and platform support. All authors participated in the discussion and wrote the manuscript.

## Competing interests

The authors declare no competing interests.
