## [Transparent Peer Review file · Nature Communications]

NMR-Solver: Automated Structure Elucidation via Large-Scale Spectral Matching and Physics-Guided Fragment Optimization

Corresponding Author: Professor Weinan E

Version 0:

Reviewer comments:

Reviewer #1

(Remarks to the Author)

The paper presents NMR-Solver, a new computational method for the elucidation of small molecule structures on the basis of 1D ¹H and ¹³C NMR spectra. NMR-based structure elucidation is of high practical importance. The method is new (not a variant of existing methods) and results presented in the paper suggest that it constitutes a significant improvement over existing methods. The paper is written clearly and concisely.

I recommend publication after minor revision to address the following points:

Lines 181-183: Report computation times and resources needed. From the description of the NMR-Solver algorithm in the paper, one would not expect it to have computational resource requirements that would make it unfeasible to evaluate more than 1000 molecules.

Table 1: It is stated that results for the earlier methods are reported directly from the corresponding publications. Were these obtained for the same 1000 molecules on which NMR-Solver was run? For a meaningful comparison, the methods must be run on the same input data.

Table 1: Explain what is meant by Conditions: Formula. If it corresponds to the number of atoms of each element, how important is its exact knowledge for the quality of the results?

Lines 179-180: Can NMR-Solver in principle use J-coupling or multiplicity information? Although these types of information can in general not be obtained for every peak, it is clear that they should help even if they are available only for a subset of the peaks.

Line 208-209: Manual curation to remove peaks from solvents or impurities may require much human time. For comparison, authors should also report results obtained without any manual clean-up. In general, it is important to discuss the susceptibility of the results on the presence of additional artifact peaks.

Lines 281-284: "goal-directed search process, where candidate quality improves systematically at each step": Any meaningful search process has a goal, and, on the other hand, an improvement at each step is not a requirement for a successful optimization method. In contrast to the precise language used elsewhere in the paper, this sentence appears as advertisement talk, for which there is no need given the convincing results.

Lines 330-331: Could this be quantified, e.g. by running tests with a pseudo chemical shift predictor that yields the chemical shifts with predefined accuracy?

Lines 366-379: In practice, getting peak information from the experimental spectra is an important part of an automated procedure. More details should be given on how this has been done here.

Lines 505-513: Data availability: The data set for ~450 reactant-product pairs compiled from JACS articles is important for

verification of the results of this paper and must be made available. It will also be useful for the evaluation of future or alternative NMR-based structure evaluation methods.

(Remarks on code availability)

Reviewer #2

(Remarks to the Author)

The paper presents a cross-modal optimization framework for inferring candidate molecular structures from corresponding ^1H -NMR and/or ^{13}C -NMR spectra. The workflow integrates two main components: (1) construction of a large-scale database of molecules with predicted spectra, and (2) a fragment-based optimization algorithm that iteratively refines molecular candidates toward a target experimental spectrum.

A comprehensive database (SimNMR-PubChem) is first generated by predicting NMR spectra for millions of PubChem molecules using NMRNet, a neural network that predicts NMR chemical shifts. The predicted spectra are smoothed via Gaussian convolution and converted into vector embeddings for both ^1H and ^{13}C NMR. Similarity between spectra is assessed through vector- and set-based distance metrics. Given a query spectrum (eventually experimental), the method retrieves an initial pool of candidate molecules whose predicted spectra best match the target.

The paper further introduces Fragment-NMR-Based Molecular Optimization (FB-MO), which fragments each candidate molecule into chemically valid substructures inheriting local spectral embeddings. New molecular candidates are generated by recombining fragments whose additive spectral vectors minimize the distance to the target spectrum. Promising structures are refined through direct NMR predictions (via NMRNet), and the candidate pool evolves iteratively toward improved spectral similarity.

Overall, I believe the work offers an interesting extension of database-driven spectrum-to-structure learning. It expands exploration beyond static molecular databases without relying on fully generative models, which can be more complex and less interpretable.

Below, I list a few points that I believe could benefit from clarification or discussion:

(p. 4) It is unclear whether the authors rely on the pre-trained NMRNet model from the original publication or have re-trained/fine-tuned it on the present dataset. I think that clarifying this is important for assessing the comparison across baselines and the reproducibility of the reported performance.

(p.13) The choice to represent spectra as unordered multisets of annotated peaks yields a compact and 'practical' feature space for machine learning. However, it seems to me that this representation assumes extensive pre-processing (e.g., peak-picking, integration, multiplicity assignment via MestReNova), which may limit scalability to raw experimental data or spectra acquired under diverse conditions. I believe the authors should discuss how 'robust' their approach is to differences in acquisition or pre-processing pipelines.

(p. 6-7), Results are compared to baselines by reporting performance numbers directly from the respective papers. However, it is also stated in the main text that results are based on 1,000 random test molecules. It is unclear whether different models are evaluated on the same test set. Additionally, the "Formula" entry under the "Conditions" column appears in every row. Does this imply that all baselines use prior knowledge of the molecular formula? Clarification is needed to ensure fair and rigorous comparison.

(p. 16-17) The fragmentation and recombination process follows predefined heuristics that modify atoms only at cleavage sites. Are alternative fragmentation schemes possible or considered?

(p. 17) From the main text, it appears that recombination operates mainly on fragment pairs (F_1 , F_2). Can the approach be extended to combinations of more than two fragments, and if so, how would the spectral inheritance and scoring generalize to that case?

In summary, I believe this paper presents an interesting idea bridging molecular databases, NMR prediction, and fragment-based molecular optimization. With further clarifications I think the work could provide a valuable contribution to machine learning approaches for NMR-based structure elucidation.

(Remarks on code availability)

The code in general is well structured and documented. However, there are some areas which deviate from standard Python package design. For example, there is no requirements file if users would like to install the code outside of the provided Docker image. There also isn't a setup.py/pyproject.toml file which is necessary to make the library easily installable. These should be added for improved useability.

Reviewer #3

(Remarks to the Author)

I co-reviewed this manuscript with one of the reviewers who provided the listed reports. This is part of the Nature

Communications initiative to facilitate training in peer review and to provide appropriate recognition for Early Career Researchers who co-review manuscripts.

(Remarks on code availability)

Reviewer #4

(Remarks to the Author)

The paper „R-Solver: Automated Structure Elucidation¹ via Large-Scale Spectral Matching and² Physics-Guided Fragment Optimization” by Jin et al. describes a software for molecular structure elucidation based on ¹H and ¹³C NMR spectra. It can also perform database search. The task is very ambitious and the authors approach it in a novel way. The paper and the software are very easy to follow and engaging. I tested the program using Bohrium Apps on four compounds: two from SDBS data base (2642 and 1574) and two from HMDB (0000190 and 0000042), and the results are:

1. Database search works quite well, I got:
 - sdb 2642 – perfect (first on the list)
 - sdb 1574 – fourth on the list
 - hmd 0000190 – first on the list, but ionized
 - hmd 000042 – fail
2. Structure elucidation failed in all cases except for lactic acid.

By the way, it would be more convenient to have the peak lists preserved when switching tabs from Database Search to Structure Elucidation.

The results were slightly disappointing, but it is impossible to solve the problem of structure elucidation based solely on 1D spectra. Still, however, the program is far more useful in database search than everything I saw on the „NMR market”. Therefore, I would recommend publishing this paper as it stands and making the software widely available. I am sure that it can be much improved as it becomes widely used. I will certainly use it.

(Remarks on code availability)

Code is ok. I tested the app using Bohrium website, as suggested on GitHub. It was very useful!

Version 1:

Reviewer comments:

Reviewer #1

(Remarks to the Author)

The authors have carefully addressed the remarks made by the reviewers and have substantially improved the paper. I recommend publication of the revised manuscript.

(Remarks on code availability)

Reviewer #2

(Remarks to the Author)

(Remarks on code availability)

Reviewer #3

(Remarks to the Author)

(Remarks on code availability)

Reviewer #4

(Remarks to the Author)

The Authors identified a discrepancy between their results and my tests, which they attributed to the presence of TMS peaks. I have no further comments and recommend the paper for publication.

(Remarks on code availability)

Response to Reviewers

Manuscript ID: NCOMMS-25-80993

NMR-Solver: Automated Structure Elucidation via Large-Scale Spectral Matching and Physics-Guided Fragment Optimization

Yongqi Jin, Jun-Jie Wang, Fanjie Xu, Xiaohong Ji, Zhifeng Gao, Linfeng Zhang, Guolin Ke*, Rong Zhu*, Weinan E*

Dear Reviewers,

We thank the reviewers for their careful reading of our manuscript and for the constructive comments. Below we list each comment followed by our response.

Response to Reviewer #1

Comment 1. Lines 181-183: Report computation times and resources needed. From the description of the NMR-Solver algorithm in the paper, one would not expect it to have computational resource requirements that would make it unfeasible to evaluate more than 1000 molecules.

Response: We thank the reviewer for this suggestion. We have added a subsection 4.7 summarizing the computational resources and average runtime per molecule for NMR-Solver:

All experiments were conducted on a single NVIDIA GeForce RTX 4090 GPU. Under the default algorithmic settings, the average wall-clock time per molecule is 94 s when element-based screening is enabled (i.e., when the elements or molecular formula of the target molecule are provided), and 106 s when element-based screening is disabled.

(Revised manuscript: Section 4.7, lines 513–517.)

This indicates that the method is computationally feasible for datasets of 1,000 molecules or larger. In practical deployments, the method naturally supports multi-GPU and multi-process execution, allowing straightforward parallelization across molecules.

Comment 2. Table 1: It is stated that results for the earlier methods are reported directly from the corresponding publications. Were these obtained for the same 1000 molecules on which NMR-Solver was run? For a meaningful comparison, the methods must be run on the same input data.

Response: We thank the reviewer for raising this important point regarding the fairness of the comparison.

To address this point, we re-implemented and re-evaluated the NMR-to-Structure baseline on the same set of 1,000 randomly sampled molecules used for NMR-Solver, ensuring a consistent input dataset for direct comparison between these two methods.

For GraphGA with Multimodal Embeddings, which relies on additional IR spectra as input, we did not re-implement or re-evaluate this method on our sampled subset. Instead, we report the performance numbers directly from the original publication, as indicated in parentheses, and include them for reference only. We now clarify this distinction explicitly

in the table caption.

These revisions ensure that all direct comparisons are performed on identical input data, while results obtained under different input modalities or evaluation settings are clearly marked as reference values.

The corresponding results in Table 1 have been updated accordingly:

Table 1: Comparison with current methods on the simulated dataset. The performance gap in the ^1H -only setting occurs due to previous methods’ dependence on idealized simulated ^1H multiplicity patterns and J -coupling values. All methods are re-implemented and tested on 1,000 randomly sampled molecules. **Bold** results indicate the best performance. Numbers in parentheses indicate the results directly reported in the original publications under their respective evaluation settings, and are provided for reference only. “Conditions: Formula” refers to the known molecular formula of the target molecule, i.e., the number of atoms of each element.

Input Spectra	Conditions	Top-1 (%)	Top-5 (%)	Top-10 (%)
GraphGA with Multimodal Embeddings				
^1H NMR + ^{13}C NMR + IR	Formula	(76.02)	(87.81)	(88.91)
NMR-to-Structure				
^1H NMR	Formula	62.70 (55.32)	75.40 (73.59)	80.80 (76.74)
^{13}C NMR	Formula	48.40 (53.91)	61.70 (73.45)	70.10 (77.72)
^1H NMR + ^{13}C NMR	Formula	69.60 (66.99)	81.70 (84.09)	86.80 (86.59)
NMR-Solver (Ours)				
^1H NMR	Formula	23.10	36.20	39.30
^{13}C NMR	Formula	62.20	77.60	79.10
^1H NMR + ^{13}C NMR	Formula	66.90	87.20	89.90

(Revised manuscript: Section 2.2, Table 1.)

Comment 3. Table 1: Explain what is meant by Conditions: Formula. If it corresponds to the number of atoms of each element, how important is its exact knowledge for the quality of the results?

Response: We thank the reviewer for the question regarding the meaning and importance of “Conditions: Formula” in Table 1.

“Conditions: Formula” refers to the known molecular formula of the target molecule, i.e., the number of atoms of each element.

(Revised manuscript: Section 2.2, Table 1.)

This definition has now been clarified in the caption of Table 1 in the revised manuscript.

In NMR-Solver, this information is utilized in two ways: (1) *Element-based screening*, which restricts the search space by excluding fragments containing elements not present in the molecular formula, thereby reducing computational cost. (2) *Final constraint checking*, where candidate molecules are filtered against the exact molecular formula to improve prediction accuracy.

A detailed description of these steps is provided in Section 4.6 (second paragraph):

“When the permissible elemental composition is known, fragments containing excluded elements are filtered out at each iteration, which helps reduce computational overhead. If a molecular formula is specified, candidate molecules are then evaluated against this constraint during the final stage.”

The impact of these molecular-formula and element-type constraints on prediction quality is analyzed experimentally in Section 2.3, with corresponding results reported in Supplementary Tables 4 and 5.

Comment 4. Lines 179-180: Can NMR-Solver in principle use J-coupling or multiplicity information? Although these types of information can in general not be obtained for every peak, it is clear that they should help even if they are available only for a subset of the peaks.

Response: We thank the reviewer for the insightful question.

In principle, NMR-Solver can incorporate additional NMR observables beyond chemical shifts. The core design of the framework relies on comparing simulated spectra of candidate molecules with the experimental spectrum and optimizing molecular structures based on their discrepancies. This comparison can, in theory, utilize chemical shifts, multiplicity patterns, and J-coupling information.

In the original manuscript, we briefly described an algorithmic approach for incorporating multiplicity information into NMR-Solver (Supplementary Note 3). In the revised manuscript, we additionally present experimental results using a rule-based ^1H multiplicity predictor. These results show a modest improvement over using chemical shifts alone (Supplementary Table 10):

A comparison between using multiplicity patterns in the ^1H NMR spectra and using chemical shifts alone is provided in Supplementary Table 10. Incorporating multiplicity yields a modest yet consistent improvement. Multiplicity and J-coupling information are expected to offer further gains for NMR-based structure elucidation as more accurate prediction methods become available.

(Revised manuscript: Supplementary Note 3)

Supplementary Table 10. Comparison of NMR-Solver performance with or without multiplicity information. Cells shaded in gray indicate improved performance over using chemical shifts alone.

NMR Type	Condition	Top-1(%)	Top-3(%)	Top-10(%)	Tani@1	Tani@3	Tani@10
NMR-Solver (chemical shifts only)							
^1H	No	0.67	1.11	3.78	0.196	0.244	0.294
	Elements	3.11	7.33	12.67	0.269	0.342	0.410
	Formula	20.00	27.33	31.78	0.382	0.445	0.488
NMR-Solver (multiplicity included)							
^1H	No	1.60	2.29	5.03	0.224	0.266	0.319
	Elements	3.78	8.22	11.56	0.293	0.353	0.420
	Formula	18.44	25.78	32.00	0.382	0.449	0.493

(Revised manuscript: Supplementary Table 10)

The limited improvement can be attributed to several factors: (1) peak overlap and annotation ambiguities (e.g., peaks labeled as 'm'), which make rule-based multiplicity assignment unreliable. As a result, only the simplest multiplicity patterns can be exploited, while such simple patterns typically carry limited structural information; and (2) occasional mislabeling in experimental spectra, for example, two doublets ('d') incorrectly annotated as a doublet of doublets ('dd'), whereas chemical shifts are generally more robust to such annotation errors.

Regarding J -coupling constants, their practical use is further complicated by the strong dependence on molecular conformation, particularly for flexible molecules. In the absence of explicit conformational ensemble calculations, there is currently no reliable way to obtain accurate J -coupling values, which makes it difficult to encode this information using simple rule-based schemes.

We believe that multiplicity and J -coupling information nevertheless have the potential to further strengthen NMR-based structure elucidation, particularly as more accurate raw-spectrum parsing methods and improved predictive tools for these observables become available.

Comment 5. Line 208-209: Manual curation to remove peaks from solvents or impurities may require much human time. For comparison, authors should also report results obtained without any manual clean-up. In general, it is important to discuss the susceptibility of the results on the presence of additional artifact peaks.

Response: We thank the reviewer for raising this important point.

Peaks arising from solvents or common impurities were removed using an automated filtering script. This procedure is a standard component of NMR preprocessing pipelines and does not require manual intervention, thereby preserving scalability. In conventional manual NMR processing, the identification and removal of residual solvent signals (for example, water or residual petroleum ether) constitute one of the most critical and subjective steps. We observed that, in some literature-reported spectra, solvent-related peaks can be mistakenly annotated as product signals, introducing avoidable noise into downstream analysis. To standardize and systematize the identification of residual solvent peaks, we therefore adopted an automated filtering strategy. Manual inspection was conducted solely to validate the automated procedure and to ensure that the resulting curated dataset constitutes a reliable benchmark, which remains scarce in the literature.

For completeness, we provide the implementation at https://github.com/YongqiJin/NMR-Solver/blob/main/src/utils/remove_solvent.py.

Regarding the susceptibility of NMR-Solver to additional artifact peaks, only a small fraction of the 450 experimental spectra contained solvent-related peaks, making a direct comparison on the full dataset unrepresentative. To assess robustness, we conducted a controlled experiment within the spectrum-matching module by injecting synthetic "artifact peaks" or deleting peaks into clean spectra. The results, reported in Supplementary Tables 8 and 9, illustrate how the presence of extra or missing peaks affects ranking performance.

Below is a summary of the impact of random peak insertion and deletion on the spectrum-matching module, where p denotes the probability of randomly inserting or deleting a peak:

NMR Type	Top-1 Recall (%)			Top-10 Recall (%)		
	$p = 0.0$	$p = 0.1$	$p = 0.2$	$p = 0.0$	$p = 0.1$	$p = 0.2$
^1H	100	94.2	88.3	100	97.0	94.0
^{13}C	100	95.4	90.7	100	98.7	97.5
$^1\text{H} + ^{13}\text{C}$	100	99.2	98.3	100	99.9	99.9

(Revised manuscript: Summary from Supplementary Table 8 and Table 9.)

The experimental results demonstrate high robustness to extra or missing peaks, particularly when both ^1H and ^{13}C spectra are used jointly.

Comment 6. Lines 281-284: "goal-directed search process, where candidate quality improves systematically at each step": Any meaningful search process has a goal, and, on the other hand, an improvement at each step is not a requirement for a successful optimization method. In contrast to the precise language used elsewhere in the paper, this sentence appears as advertisement talk, for which there is no need given the convincing results.

Response: We thank the reviewer for the comment. We have removed the last paragraph because it largely repeats the point already made in the preceding sentence:

By prioritizing molecular structures that better explain the observed spectra, the method identifies high-quality candidates early, without the need for exhaustive exploration. In contrast, conventional approaches, such as genetic algorithms, rely on random crossover and mutation operations, lacking directed guidance.

(Revised manuscript: Section 2.5, lines 281–284.)

This revision avoids redundancy while preserving the intended comparison between NMR-Solver and conventional methods.

Comment 7. Lines 330-331: Could this be quantified, e.g. by running tests with a pseudo chemical shift predictor that yields the chemical shifts with predefined accuracy?

Response: We thank the reviewer for the insightful suggestion regarding quantifying the effect of chemical shift prediction accuracy on NMR-Solver. Direct testing with a pseudo predictor in de novo structure generation is challenging, as candidate molecules lack ground-truth experimental spectra. Instead, we approximate this effect via NMR-based molecular retrieval.

Specifically, since spectral matching is central to NMR-Solver, we emulate reduced chemical shift accuracy by adding controlled Gaussian noise to simulated spectra, effectively acting as a pseudo predictor with predefined error levels. Retrieval experiments across different noise levels were conducted, and top- k recall was evaluated. The full results are provided in Supplementary Tables 8 and 9.

Below is a summary of the impact of different noise levels on the search algorithm, σ denotes the noise level used to simulate a pseudo chemical shift predictor (σ ppm for ^1H NMR and 10σ ppm for ^{13}C NMR):

NMR Type	Top-1 Recall (%)			Top-10 Recall (%)		
	$\sigma = 0.0$	$\sigma = 0.1$	$\sigma = 0.2$	$\sigma = 0.0$	$\sigma = 0.1$	$\sigma = 0.2$
^1H	100	76.9	31.1	100	92.8	53.0
^{13}C	100	87.9	50.9	100	99.0	70.7
$^1\text{H} + ^{13}\text{C}$	100	95.5	86.2	100	99.9	95.8

(Revised manuscript: Summary from Supplementary Table 8 and Table 9.)

These results provide a quantitative view of how prediction accuracy influences downstream structure elucidation. When the prediction errors exceed approximately 0.1 ppm for ^1H shifts and 1 ppm for ^{13}C shifts, the performance of spectrum-matching deteriorates noticeably, especially in terms of top-1 recall. The current predictor used in NMR-Solver, NMRNet, achieves average errors of 0.18 ppm for ^1H and 1.1 ppm for ^{13}C on the public benchmark nmrshiftdb2. Therefore, further improving the accuracy of chemical-shift prediction models is expected to substantially enhance both NMR-based retrieval and structure elucidation accuracy.

Comment 8. Lines 366-379: In practice, getting peak information from the experimental spectra is an important part of an automated procedure. More details should be given on how this has been done here.

Response: We thank the reviewer for this valuable comment.

Because our data originate from two distinct sources and formats, we employ two corresponding extraction procedures. For literature-reported spectra (text-based formats; Sections 2.3 and 2.4), peak information is extracted using regular-expression parsing. The exact implementation is available in our open-source code: https://github.com/YongqiJin/NMR-Solver/blob/main/src/utils/nmr_parser.py.

For experimentally acquired spectra (Section 2.5), standard NMR preprocessing procedures are applied. Specifically, raw FID files are processed using conventional NMR software such as MestReNova, including baseline correction, peak picking, integration, and multiplicity assignment.

We have expanded Section 4.1 to describe these steps in greater detail.

For literature-reported spectra, we extract peak information using regular-expression parsing. For experimentally acquired spectra, peaks are annotated using standard NMR preprocessing procedures. Specifically, the raw FID files are processed using conventional NMR software such as MestReNova, including baseline correction, peak picking, peak picking, integration and multiplicity assignment.

(Revised manuscript: Section 4.1, lines 370–377.)

Comment 9. Lines 505-513: Data availability: The data set for 450 reactant-product pairs compiled from JACS articles is important for verification of the results of this paper and must be made available. It will also be useful for the evaluation of future or alternative NMR-based structure evaluation methods.

Response: We agree with the reviewer and appreciate the importance of making this dataset accessible. The dataset comprising approximately 450 reactant-product pairs compiled from JACS articles has been made publicly available. It can be accessed via GitHub: <https://github.com/YongqiJin/NMR-Solver/blob/main/data/experiment/test.txt> and is also archived on Zenodo: <https://zenodo.org/records/16952024>.

(Revised manuscript: Data availability section.)

Response to Reviewer #2

Comment 10. (p. 4) It is unclear whether the authors rely on the pre-trained NMRNet model from the original publication or have re-trained/fine-tuned it on the present dataset. I think that clarifying this is important for assessing the comparison across baselines and the reproducibility of the reported performance.

Response: Thank the reviewer for the helpful comment. We have clarified in Section 2.1 that we use the original pre-trained NMRNet model without any additional fine-tuning on our datasets, in order to avoid potential overfitting that could bias the fair comparison across baselines. To ensure full reproducibility, we have also released the exact model weights used in our experiments at <https://zenodo.org/records/16952024>.

To support rapid spectral evaluation during optimization, we employ pre-trained NMRNet, ...

(Revised manuscript: Section 2.1, line 135.)

Comment 11. (p.13) The choice to represent spectra as unordered multisets of annotated peaks yields a compact and ‘practical’ feature space for machine learning. However, it seems to me that this representation assumes extensive pre-processing (e.g., peak-picking, integration, multiplicity assignment via MestReNova), which may limit scalability to raw experimental data or spectra acquired under diverse conditions. I believe the authors should discuss how ‘robust’ their approach is to differences in acquisition or pre-processing pipelines.

Response: We thank the reviewer for the insightful comment regarding the reliance on spectral pre-processing and its implications for scalability and robustness.

We agree that representing spectra as unordered multisets of annotated peaks assumes a pre-processing stage, such as peak picking, integration, and multiplicity assignment. Our design choice follows current mainstream NMR analysis practice, where pre-processing is widely adopted because it offers interpretability, transparency, and user control, particularly for handling solvent peaks, artifacts, and ambiguous assignments. While fully end-to-end spectrum parsing methods offer greater automation, we deliberately adopt a pre-processing-based representation to enable more controllable and practical use under real experimental settings.

Importantly, NMR-Solver is explicitly designed to be robust to variations in acquisition conditions and pre-processing pipelines. As highlighted in the revised Section 4.3, the core spectrum-matching module relies on set-based similarity rather than strict vector alignment:

Compared to vector-based similarity metrics, set-based similarity is more tolerant to missing or noisy peaks, enabling final robust identification of molecules even when spectra partially overlap. This reflects a design that is inherently robust to noise and imperfections in peak extraction under realistic experimental conditions. Additional experimental evidence demonstrating these advantages is provided in Supplementary Note 4.

To quantitatively evaluate robustness, we conducted controlled experiments within the spectrum-matching module by simulating realistic experimental uncertainties, including random noise and random peak insertion or deletion. We compared set-based similarity with vector-based similarity and traditional Wasserstein distance under these perturbations. The results demonstrate that the set-based formulation used in NMR-Solver is substantially more robust to pre-processing variability.

Detailed experimental results are provided in Supplementary Note 4 and Supplementary Tables 8 and 9.

Comment 12. (p. 6-7) Results are compared to baselines by reporting performance numbers directly from the respective papers. However, it is also stated in the main text that results are based on 1,000 random test molecules. It is unclear whether different models are evaluated on the same test set. Additionally, the “Formula” entry under the “Conditions” column appears in every row. Does this imply that all baselines use prior knowledge of the molecular formula? Clarification is needed to ensure fair and rigorous comparison.

Response: We thank the reviewer for raising this important point regarding fair and rigorous comparison.

To address this point, we re-implemented and re-evaluated the NMR-to-Structure baseline on the same set of 1,000 randomly sampled molecules used for NMR-Solver, ensuring a consistent input dataset for direct comparison between these two methods.

For GraphGA with Multimodal Embeddings, which relies on additional IR spectra as input, we did not re-implement or re-evaluate this method on our sampled subset. Instead, we report the performance numbers directly from the original publication, as indicated in parentheses, and include them for reference only. We now clarify this distinction explicitly in the table caption.

These revisions ensure that all direct comparisons are performed on identical input data, while results obtained under different input modalities or evaluation settings are clearly marked as reference values.

The corresponding results in Table 1 have been updated accordingly:

Table 1: Comparison with current methods on the simulated dataset. The performance gap in the ^1H -only setting occurs due to previous methods’ dependence on idealized simulated ^1H multiplicity patterns and J -coupling values. All methods are re-implemented and tested on 1,000 randomly sampled molecules. **Bold** results indicate the best performance. Numbers in parentheses indicate the results directly reported in the original publications under their respective evaluation settings, and are provided for reference only. “Conditions: Formula” refers to the known molecular formula of the target molecule, i.e., the number of atoms of each element.

Input Spectra	Conditions	Top-1 (%)	Top-5 (%)	Top-10 (%)
GraphGA with Multimodal Embeddings				
^1H NMR + ^{13}C NMR + IR	Formula	(76.02)	(87.81)	(88.91)
NMR-to-Structure				
^1H NMR	Formula	62.70 (55.32)	75.40 (73.59)	80.80 (76.74)
^{13}C NMR	Formula	48.40 (53.91)	61.70 (73.45)	70.10 (77.72)
^1H NMR + ^{13}C NMR	Formula	69.60 (66.99)	81.70 (84.09)	86.80 (86.59)
NMR-Solver (Ours)				
^1H NMR	Formula	23.10	36.20	39.30
^{13}C NMR	Formula	62.20	77.60	79.10
^1H NMR + ^{13}C NMR	Formula	66.90	87.20	89.90

(Revised manuscript: Section 2.2, Table 1.)

Regarding the use of the molecular formula as prior information, we ensured that all baseline results incorporate this knowledge consistently with the original publications. (The inclusion of this column in Table 1 is aligned with the experiments in Section 2.3, where we analyze the impact of molecular formula and elemental composition on the quality of structure elucidation, with results reported in Supplementary Tables 4 and 5.)

Comment 13. (p. 16-17) The fragmentation and recombination process follows predefined heuristics that modify atoms only at cleavage sites. Are alternative fragmentation schemes possible or considered?

Response: We thank the reviewer for this insightful comment.

The original sentence,

"During the molecular fragmentation and recombination process to generate new molecules, atoms at the cleavage sites of the fragments are the only ones that undergo significant changes, while the remaining atoms maintain relatively stable local atomic environments,"

was somewhat vague. We have revised it to:

During the molecular fragmentation and recombination process to generate new molecules, atoms at the cleavage sites of the fragments are the only ones whose local environments undergo substantial changes, while the local environments of the remaining atoms remain relatively stable.

(Revised manuscript: Section 4.5, lines 469–473.)

This revised wording more precisely describes how the local atomic environments change

during fragmentation and recombination, which reflects the variations in chemical shifts and directly affects the accuracy of predicting the chemical shifts of Child molecules based on the Parent molecules.

Using all possible fragmentation schemes is feasible and has also been explored. According to our former experiments, employing all possible fragmentations results in a modest decrease in performance, because the chemical shift prediction error near the cleavage sites tends to be larger. Therefore, we design the fragmentation and recombination rules under the principle that

*“These constraints ensure that the circular neighborhood of **radius 1** for all carbon atoms remains unchanged,” (Section 4.5, lines 479–480)*

to achieve a **trade-off** between accurate chemical shift prediction for newly generated molecules and the diversity of generated chemical space.

A simple schematic is provided to illustrate the rationale behind this trade-off:

Comment 14. (p. 17) From the main text, it appears that recombination operates mainly on fragment pairs (F_1, F_2) . Can the approach be extended to combinations of more than two fragments, and if so, how would the spectral inheritance and scoring generalize to that case?

Response: We thank the reviewer for this insightful question.

As the reviewer noted, in each iteration we consider only pairwise combinations of fragments from the molecular pool (i.e., (F_1, F_2)). The final assembly of multiple fragments is achieved through multiple rounds of iterative optimization.

Restricting each iteration to pairwise recombination offers a key advantage. It enables efficient filtering of potential fragment recombinations using the L^2 -distance

$$\|\mathbf{v}_{\text{target}} - \mathbf{v}_{F_1} - \mathbf{v}_{F_2}\|_2,$$

which can be further accelerated via vector-database indexing techniques, greatly improving computational efficiency (Section 4.5, lines 488–490).

Extending the approach to direct recombination of more than two fragments within a single

iteration would render such indexing-based filtering infeasible, as enumerating all possible fragment combinations becomes computationally intractable.

Comment 15. The code in general is well structured and documented. However, there are some areas which deviate from standard Python package design. For example, there is no requirements file if users would like to install the code outside of the provided Docker image. There also isn't a `setup.py/pyproject.toml` file which is necessary to make the library easily installable. These should be added for improved useability.

Response: We thank the reviewer for the positive feedback on our open-source code and for the constructive suggestions regarding package design.

A `requirements.txt` has been updated, listing all standard open-source dependencies to facilitate installation in a standard Python environment (<https://github.com/YongqiJin/NMR-Solver/blob/main/requirements.txt>). The only exception is a modified version of the `unimol-tools` library (<https://pypi.org/project/unimol-tools/>), which is currently not publicly available and requires the Docker image to run.

To ensure a stable and deployable environment and to avoid the complexity of managing package versions, we provide a unified Docker image containing all necessary dependencies. Because the code relies on this Docker-based environment, creating a `setup.py` or `pyproject.toml` for direct pip installation is not particularly meaningful at this stage. Once the modified `unimol-tools` is publicly released, we plan to provide a standard pip-installable package to further improve usability.

Response to Reviewer #3

Comment 16. I co-reviewed this manuscript with one of the reviewers who provided the listed reports. This is part of the Nature Communications initiative to facilitate training in peer review and to provide appropriate recognition for Early Career Researchers who co-review manuscripts.

Response: We sincerely thank the reviewer for their contribution to the peer review process. No changes to the manuscript were required in response to this comment.

Response to Reviewer #4

Comment 17. The paper “NMR-Solver: Automated Structure Elucidation via Large-Scale Spectral Matching and Physics-Guided Fragment Optimization” by Jin et al. describes a software for molecular structure elucidation based on ^1H and ^{13}C NMR spectra. It can also perform database search. The task is very ambitious and the authors approach it in a novel way. The paper and the software are very easy to follow and engaging. I tested the program using Bohrium Apps on four compounds: two from SDBS data base (2642 and 1574) and two from HMDB (0000190 and 0000042), and the results are:

1. Database search works quite well, I got:

- sdb 2642 – perfect (first on the list)
- sdb 1574 – fourth on the list
- hmdb 0000190 – first on the list, but ionized
- hmdb 000042 – fail

2. Structure elucidation failed in all cases except for lactic acid.

By the way, it would be more convenient to have the peak lists preserved when switching tabs from Database Search to Structure Elucidation.

The results were slightly disappointing, but it is impossible to solve the problem of structure elucidation based solely on 1D spectra. Still, however, the program is far more useful in database search than everything I saw on the “NMR market”. Therefore, I would recommend publishing this paper as it stands and making the software widely available. I am sure that it can be much improved as it becomes widely used. I will certainly use it.

Response: We sincerely thank the reviewer for the detailed testing and thoughtful feedback. We are glad that the reviewer found the software easy to use and useful, particularly for database searching.

We reproduced the four examples mentioned by the reviewer from the SDBS database (<https://sdb.sdb.aist.go.jp/SearchInformation.aspx>) and the HMDB database (<https://www.hmdb.ca/textquery>). Detailed information for these examples is summarized below. Peaks near 0 ppm were excluded in HMDB0000190 and HMDB0000042, as they correspond to the TMS internal standard used for calibration.

- **sdb 2642: Pyruvic acid**

Structure: CC(=O)C(=O)O

Formula: $\text{C}_3\text{H}_4\text{O}_3$

^{13}C NMR (ppm): 193.67, 161.37, 25.45

^1H NMR (ppm): 8.70 (1H), 2.534 (3H)

NMR text for NMR-Solver: ^1H NMR 8.70 (1H), 2.534 (3H); ^{13}C NMR 193.67, 161.37, 25.45.

- **sdb 1574: Hexanenitrile**

Structure: CCCCCC#N

Formula: $\text{C}_6\text{H}_{11}\text{N}$

^{13}C NMR (ppm): 119.90, 30.87, 25.21, 21.96, 17.11, 13.78

^1H NMR (ppm): 2.33 (2H), 1.67 (2H), 1.72 (4H), 0.94 (3H)

NMR text for NMR-Solver: ^1H NMR 2.33 (2H), 1.67 (2H), 1.72 (4H), 0.94 (3H); ^{13}C NMR 119.90, 30.87, 25.21, 21.96, 17.11, 13.78.

- **HMDB0000190**

Structure: C[C@H](O)C(O)=O

Formula: C3H6O3

¹³C NMR (ppm): 185.0763, 71.2486, 22.8053, ~~-0.0013~~ (TMS)

¹H NMR (ppm): 1.32 (4H, d, *J* = 3 Hz), 4.10 (2H, q, *J* = 6.93 Hz)

NMR text for NMR-Solver: 1H NMR 1.32 (4H, d, *J* = 3 Hz), 4.10 (2H, q, *J* = 6.93 Hz); 13C NMR 185.0763, 71.2486, 22.8053.

- **HMDB0000042**

Structure: CC(O)=O

Formula: C2H4O2

¹³C NMR (ppm): 184.1082, 25.9762, ~~0.0018~~ (TMS)

¹H NMR (ppm): 1.91 (3H)

NMR text for NMR-Solver: 1H NMR 1.91 (3H); 13C NMR 184.1082, 25.9762.

The results are summarized below. All results were obtained using the NMR-Solver Web App with default settings (elements C, H, N, O) and using only the NMR text as input.

No.	Rank		
	DB Search	SE (before EF)	SE (after EF)
sdbs 2642	1	39	1
sdbs 1574	1	1	1
HMDB0000190	not found	2	1
HMDB0000042	not found	16	1

Notes: DB Search = Database Search; SE = Structure Elucidation; EF = Elemental Filter. Ranks indicate the position of the correct structure in the output list (1 = top rank).

The screenshot below shows the structure elucidation results generated by NMR-Solver:

- sdbs 2642 (after elemental filter):

- sdbs 1574 (before elemental filter):

- HMDB0000190 (before elemental filter):

- HMDB0000042 (after elemental filter):

We think the discrepancy between our results and the reviewer's tests may arise from the handling of TMS peaks. For these examples, which have a small number of atoms and peaks, the inclusion of extraneous peaks can have a disproportionately large effect.

Nevertheless, the third and fourth examples demonstrate that NMR-Solver exhibits a de-

gree of robustness to spectral noise. For instance, HMDB0000042 contains no labile protons in the ^1H spectrum, and HMDB0000190 records labile protons that deviate substantially from the model predictions. Despite these challenges, NMR-Solver successfully solved both structures when the elemental composition was provided.

We also appreciate the reviewer's suggestion to preserve peak lists when switching between Database Search and Structure Elucidation, and will consider implementing this enhancement in future versions.

We appreciate the reviewer's recommendation for publication and their encouragement, and hope that broader use of NMR-Solver will further enhance its capabilities through user feedback and ongoing development.

Comment 18. Code is ok. I tested the app using Bohrium website, as suggested on GitHub. It was very useful!

Response: Thank the reviewer for testing the software and for the positive feedback. We will continue to refine the software based on user feedback.

We would like to once again thank the reviewers for their constructive comments. We believe that this feedback has significantly improved the manuscript.

Sincerely,

All authors